# Digital Transformation among SMEs: Does Gender Matter?

**Khorshed Alam** [1,2,*] , **Mohammad Afshar Ali** [1,3] , **Michael O. Erdiaw-Kwasie** [4] , **Peter A. Murray** [1,5] **and Retha Wiesner** [6]

1 School of Business, Faculty of Business, Education, Law and Arts, University of Southern Queensland, Toowoomba, QLD 4350, Australia; MohammadAfshar.Ali@usq.edu.au (M.A.A.); peter.murray@usq.edu.au (P.A.M.)
2 Centre for Health Research, University of Southern Queensland, Toowoomba, QLD 4350, Australia
3 Allied Health and Human Performance Unit, University of South Australia, Adelaide, SA 5000, Australia
4 Sustainable Enterprise Division, Asia Pacific College of Business and Law, Charles Darwin University, Darwin, NT 0800, Australia; Michael.erdiaw-kwasie@cdu.edu.au
5 Rural Economies Centre of Excellence, University of Southern Queensland, Toowoomba, QLD 4350, Australia
6 School of Business, Faculty of Business, Education, Law and Arts, University of Southern Queensland, Springfield Central, QLD 4300, Australia; retha.wiesner@usq.edu.au
* Correspondence: Khorshed.Alam@usq.edu.au

**Abstract:** The COVID-19 pandemic has highlighted and exacerbated some of the challenges that small and medium enterprises (SMEs) face in times of crisis, disrupting their operations, weakening their financial positions, and exposing them to a wide range of financial risks. While previous studies have viewed digital transformation as a vital source of innovation and productivity growth for economic recovery in SMEs, there has been limited focus on digital transformation in the regional context, with very little attention focused on women-led enterprises. This study aims to investigate (i) the determinants of perception of digital transformation among regional SMEs, and (ii) whether the gender of the SME owner or manager has an impact on the drivers of the digital transformation experiences of SMEs operating in regional Australia. Building upon the resource-based view, this study uses a unique dataset of 281 SMEs collected from a survey conducted within a regional area of Queensland, Australia. Employing Feasible Generalised Least Squares and Generalised Least Squares estimations, the study found that the perceptions of digital transformation can be explained by the use of social network platforms, innovation processes, workplace culture, and information and communication technologies. This study also found that there is a significant difference between female-led and male-led SMEs regarding their perceptions of digital transformation. This study offers two key policy and practical insights: (i) digital transformation of regional SMEs should be used as a fundamental tool for crisis recovery strategies, and (ii) the need for policymakers to mainstream gender into postcrisis transformative interventions and policies should be fast tracked.

**Keywords:** Australia; Blinder-Oaxaca decomposition; digital transformation; resource-based view; small and medium enterprise (SME); COVID-19

## 1. Introduction

Research has shown that small- and medium-sized enterprises (SMEs) are a critical catalyst for employment generation and economic growth in regional and rural communities [1,2]. Following the COVID-19 pandemic, new survey evidence indicates that SMEs across many countries including Australia are divided about the perceived threats and opportunities of pandemic-driven digital disruption [3]. Whereas the financial burden imposed by the pandemic and the limited digital knowledge to overcome this challenge has affected some SMEs [4], the pandemic has triggered digital innovations in other SMEs facilitating faster recovery in the post-COVID-19 era [5–7].

Nevertheless, despite the game-changing potential of digital transformation for Australian SMEs, it is unclear how SMEs in regional Australia perceive digital transformation

and associated digital disruptions, and what differences exist between female-led and male-led businesses regarding how they perceive of these disruptions. Accordingly, this study explores the perceptions of regional SME owners/managers regarding digital opportunities, and how gender inequality varies considering the determinants of digital transformation.

Following the period of closure and movement prevention policies adopted by national and state governments due to the COVID-19 pandemic, SMEs are confronted with various difficulties and challenges [4,5]. In response to these crisis-related challenges, many SMEs have adopted digital technologies that improve the effectiveness and efficiency of existing processes. Organisational performance through the exploitation of new digital opportunities is also enhanced through such technological advancements [8–10]. The rapid deployment and uptake of digital technologies in SMEs provide not only opportunities to achieve higher productivity growth but also pose risks such as a loss in market share [11] or even the threat of ceasing to exist [4,12,13]. Despite much attention to, and optimism about, the role that digital technologies can play in SMEs' economic growth and pandemic recovery processes [7,14], there is a dearth of studies on these issues within the regional Australian context.

This study reflects an emphasis on digital transformation—as distinct from similar terms such as digitalization, digitation or digital disruption, that are often used interchangeably—to refer to the process of digitising resources to enable the transformation of customer experiences, operational processes, and ultimately, the efficiency gains of a business [15,16]. It is the process of leveraging digital technologies and digitalised data to create new and revise existing business models. This often becomes imperative to meet the challenges of new market demands and changing business environments. The rapid pace of digital transformation could have a significant impact on enabling change across SMEs [17,18]. For instance, Benitez et al. [19] suggest that information technology infrastructure increases an SME's ability to acquire and share information from, and to the market, thus better enabling its ability to leverage its technical and human resources. Australian SMEs constitute about 96 percent of businesses and play a substantial role in the economy and society as well as contributing to employment [20,21]. There are several barriers to accessing and using effective digital technologies including, but not limited to, a lack of high-quality and affordable infrastructure, a lack of trust in such technologies, and the shortage of digitally skilled people [22]. Moreover, scholars have noted the lack of compatibility, technological integration, and regulatory support for digital technologies such as cloud computing [23,24]. New national infrastructure initiatives such as the National Broadband Network (NBN) create an opportunity for SMEs in Australian regional areas to access fast and affordable digital technologies [25].

With respect to the effect of gender on new technology adoption as a basis for the current study, data trends suggest that gender inequality in the digital sphere is highly prevalent across countries [26]. The emergence of opportunities and threats associated with digital technologies may create a level of gender inequality between women-led SMEs and their male counterparts. Digital inequality refers to unequal access and use of information and communication technologies [27], while gender digital divide is defined as the unequal opportunity to use and access such technologies between men and women in social, cultural, political, and economic domains [28]. In the regional context, digital transformation generates a considerable opportunity to boost SME performance. Research indicates that due to the pandemic, gender inequality is more pronounced in terms of digital access, ownership of digital devices, as well as digital fluency [29,30], which has created a perfect storm for women-led SMEs [31,32]. Research also highlights a lack of participation in technical jobs including the need to increase mentoring strategies for women at work [33].

While digital technologies provide opportunities to firms across all industries globally, the enablers of digital transformation are yet to be explored, particularly at the regional scale of development which is a significant gap. This research problem highlights the digital technology divide across regional communities not just in Australia but more

broadly [25,26], representing a second noticeable gap within extant studies. Thus, this study is both timely and important since what is at stake is a better understanding of the resources and digital technological strategies that underlie regional growth. The study is also important as it provides new evidence of the perceptions of SME owners/managers regarding digital transformation opportunities. Moreover, the study explains why different determinants of digital transformation influence male and female SME owners/managers perceptions. Given these gaps, the current study strives to answer the following research questions concerning the perspectives of regional SMEs: (i) what are the resource-based enablers of digital transformation within the Australian regional SME context? and (ii) do male and female SME owners/managers differ in their perceptions regarding the adoption of digital transformation? The findings will contribute to the emergence of the digital transformation literature by examining the variables that influence the owner/manager's perception of digital transformation in rural and regional settings.

## 2. Theoretical Construct and Development of Hypotheses

### 2.1. The Resource-Based View

The resource-based view (RBV) suggests that it is the combination of strategic resources and capabilities of a firm that leads to sustainable competitive advantage [34,35]. In the current study, we contend that the ability of a firm to integrate, build and reconfigure digital technology capabilities and combine them with other broader managerial capabilities, will help to create sustainable competitive advantage within regional SMEs [35]. In this study, the focus is on the owner/manager's perception of digital transformation to deliver sustainable competitive advantage. Managerial capabilities in this study refer to the capabilities with which managers build, integrate, and reconfigure organisational resources and competencies [36]. The combined capability of an SME to integrate, assemble, and deploy digital technology resources alongside a broader set of managerial capabilities, will help to increase performance while capitalising on business opportunities. Thus, a broader question to ask in respect of the RBV of managerial capabilities is what do they look like from an SME perspective?

According to Barney [34], a firm's capital resources should be (i) valuable (V), relative to opportunities and threats, (ii) rare (R), relative to current and potential competitors, (iii) imperfectly imitable (I), or not easily replicated, and (iv) non-substitutable, or have no equivalent substitutes (O). Tangible assets refer to all assets that create tangible value, e.g., equipment and buildings, while intangible assets refer to knowledge, information, and ideas [37,38]. A tangible asset or physical capital resource such as manufacturing processes will often be accompanied by intangible routines and knowledge which are hard to replicate [39]. Similarly, strong evidence of digital technology adoption coupled with managerial knowledge will create a combined capability which may be unique, rare and valuable. It can be argued, that when technological resources as a tangible asset such as digital technologies are combined with human resource intangible assets such as a culture that values the role of women and equal opportunity, the combination of strategic resources and capabilities may lead to a sustained competitive advantage. We later explore these strategic resources by building a composite index through modelling equations to define the level of digital transformation perceived by business owners/managers.

### 2.2. SMEs' Perception of Digital Transformation

Digital technologies create an environment in which they can be perceived as both an opportunity and a threat [40]. Perceptions of stakeholders toward digital disruption depend on the speed of response to the associated threat and how well small businesses capitalise on the opportunities. Existing research on the opportunities and risks associated with digital transformation, particularly in the SME context is limited. Hamil [41] investigated the impact of digital disruption on SMEs and identified a list of opportunities and threats that digital disruption present including but not limited to, ease of access to international markets, efficiency and effectiveness, and lack of digital technology adaptation.

Like most economies, the Australian digital economy is expected to grow rapidly in the future with a new wave of digital technologies emerging in various industries [42]. Since there is a rapid improvement in a range of digital technologies, SMEs are now faced with entirely different challenges. Access and use of digital technologies have a substantial influence on SMEs [43]. Many opportunities and challenges associated with these technologies come from the ability of SMEs to access and use digital devices [17,44,45]. SMEs use digital technologies to improve their products and services so that firm performance is enhanced [45–47]. Deloitte [48] finds that SMEs with high digital technology engagement earn twice as much revenue per employee as businesses with low engagement. The increased rate of adoption of digital technologies among SMEs in Australia may be perceived by owners/managers as an opportunity to create an environment in which they can better exploit opportunities. Therefore, digital transformation perception reflects the owner/manager's perception of adopting digital technology which can be a powerful enabler of SME performance.

### 2.3. Enablers of Digital Transformation

There are many firm-level digital technology-related resources, capabilities, and managerial competencies such as social media, web presence, E-commerce, and managerial competencies that could determine the perception of users of digital technologies. For example, SMEs' readiness and willingness to adopt these types of technology are generally referred to as information technology (IT) adoption in the IT literature. In addition, firms are in a better position to adopt IT innovations such as digital technologies when they possess an appropriate IT infrastructure [49,50]. We next discuss these digital technology-related resources and capabilities that help release digital technology innovation.

### 2.3.1. Social Media

The adoption of social media in businesses has enabled SMEs to interact with employees and customers and is viewed as an essential tool that can help create opportunities in the market [51–53]. Deloitte [47] finds that thirty-one percent of Australian businesses have a social media presence, which is used mostly for marketing and to engage with stakeholders. Businesses that use social media however have better systems to engage customers in innovation, potential employees in recruitment, and facilitate internal and external collaboration. Ainin et al. [54] found that social media use in SMEs has a strong positive impact on financial and nonfinancial performance with the key factors including compatibility, cost-effectiveness, and interaction.

### 2.3.2. Presence of Websites

The presence of a website is considered a strategic tool in a dynamic environment facilitating better sharing of business information while improving business transactions that promote the interests of a wide range of people [55,56]. Ramayah et al. [57], for example, investigated the factors influencing SME website adoption in the Malaysian context and found that SMEs with innovative owners/managers have a positive attitude towards IT adoption deriving a relative advantage. The Australian Productivity Commission [11] argues that businesses adopt websites for marketing purposes primarily to enter national and international markets. Traditional businesses may deal with digital disruption by promoting their presence more powerfully through a website thus increasing their online presence and information dissemination. Having an effective website presence enables the business to enhance customer awareness and interaction with existing and potential customers.

### 2.3.3. E-Commerce

The adoption of E-commerce has disrupted the existing retail system with new marketplaces and shopping tools, which have created market opportunities [58,59]. Previously, consumers resorted to legacy modes of commerce such as store and hardcopy advertisements in the purchasing decision. Contemporary consumers make use of other digital

devices such as Tablets, iPads, and smartphones to search for information about products and prices. Hanna et al. [60] noted that price transparency in SMEs is increasing due to the proliferation of E-commerce activities. However, despite the increased literature on SMEs and E-commerce, studies focused within a regional and rural context are limited [61]. Abebe [62] suggests that the adoption of E-commerce has a positive impact on financial performance, more so when SMEs have a higher level of entrepreneurial orientation. E-commerce competencies can be perceived by managers and owners as extremely valuable and a vital resource that can be used to improve the performance of their business.

Studies have shown that incorporating digital technology resources into a firm's stock of capabilities, e.g., managerial competencies into products, services, and processes, has important implications for how SMEs gain and sustain a competitive advantage. Based on the RBV, we argue that the ability of SMEs in regional areas to integrate, assemble and deploy enablers of digital transformation and managerial competencies such as social media, web presence, E-commerce, staff digital skills, and workforce flexibility, can bring digital opportunities to regional SMEs as a means of increasing competitive advantage. Accordingly, we hypothesise:

**Hypothesis 1 (H1).** *E-commerce, web presence, social media, and other capital resource factors enable digital transformation in regional SMEs.*

### 2.4. Gender Inequality and Digital Transformation

There is a dearth of empirical studies on the effect of gender on digital technologies [63,64]. Previous studies argue that research concerning women's use of digital technologies is largely ignored [65,66]. Wiesner [67] found that isolation, lack of like-minded peers, lack of mentors and lack of access to networks to be major barriers for women in rural enterprises. Therefore, social media can lead to greater empowerment for women because it changes the way women network and build relationships and support [68]. Digital technology not only provides women entrepreneurs with access to valuable information about their business but also facilitates their efforts to balance their work and life [69].

Social technologies in comparison to other digital technologies have provided a platform for increased communication, collaboration, and greater exposure to profit-making ventures [68] (p. 1103). However, the rate of digital technological growth could create digital disruption by elevating both opportunities and threats leading to more gender inequality among SMEs. While scholars have noted that women place more emphasis on social ties and commitment, it could also pose a significant privacy risk when sharing information [70]. It appears that the worldwide pandemic has not only increased women's access to digital technology but has also highlighted various digital fluency problems [29,30]. Digital fluency of individuals in technologically connected SMEs may help to narrow gender digital inequality pointing to a significant opportunity within rural locations. Overall, previous studies have neglected the influence of gender equality as a key driver of digital transformation which exposes an important research gap that needs to be investigated. This study addresses this gap by analysing a range of digital gender differences regarding the drivers of digital transformation. Accordingly, we hypothesise:

**Hypothesis 2 (H2).** *There is a significant difference between male and female SME owners/managers regarding their perceptions of digital transformation in regional areas?*

### 3. Materials and Methods

#### 3.1. Variables and Model

The methodology of the study including choice of methods, econometric models, variable selection, data collection and analysis plan is elaborated in this section. This includes Feasible Generalised Least Squares (FGLS) and Generalised Least Squares (GLM), which are used to explore the association between digital transformation and its determinants as set

out within the theoretical construct. In addition, to examine the interaction or moderating effect of gender, GLM has been applied. Based on the theoretical construct, the variables used in this study are summarised in Table 1 with an added variable definition providing justification for the variables used.

**Table 1.** List of variables used in the analysis.

| Variable Name | Symbol | Variable Definition |
|---|---|---|
| Perception of the extent of digital transformation | DT | A composite index was created using three statements. First, it was asked whether a business was significantly impacted and disabled by digital technologies as opposed to significantly improved and enabled by digital technologies (5-point scale from 1 = 'significantly impacted and disabled' to 5 = 'significantly improved and enabled'). Second, opportunities resulted from the use and access of digital technologies was measured on a 5-point scale (1 = 'definitely not' to 5 = 'definitely yes'). Third, the change of adoption of digital technologies in a business was measured on a 5-point scale from 1 = 'very negative' to 5 = 'very positive'. |
| E-commerce | ECO | A dummy variable indicating whether a firm has adopted E-commerce applications, i.e., using the internet for selling goods and services (1) or not (0). |
| Web presence | WSP | A dummy variable indicating whether a firm has a web presence (1) or not (0) |
| Social networking | SN | A dummy variable indicating whether a firm use any of the social networking sites (Facebook, YouTube, Twitter, WordPress, Tumblr, Blog and others) for business purposes (1) or not (0) |
| Workplace culture | CUL | An indicator reflecting the respondent's opinion that organization follows flexible work practices' (5-point scale from 1 = 'not true at all' to 5 = 'very true'). |
| Innovation index | INNOV | A composite index was developed to measure to what extent an enterprise is involved in innovation activities using PCA. It is a composite measure of four indicators reflecting an enterprise's level of engagement in the product, process, marketing and organizational innovation. Each of the innovation indicators was measured by the opinion of the respondents on respective innovation actions (5-point scale from 1 = 'greatly decreased' to 5 = 'greatly increased'). |
| Gender of the owner/manager | GENDER | A dummy variable takes value 1 if Male, and 0 otherwise. |
| Number of Employees | EMP | Natural logarithm of a number of employees. This variable is used as a proxy measure of the size of the firm. |
| Remoteness | REM | A dummy variable indicating whether a firm is located in remote/rural areas (1) or not (0). |
| Internationalization | INT | The ratio of revenue of a firm is from international export activities. |
| Profitability | PROFIT | A categorical variable indicating the level of profitability of the business enterprise. |
| ICT skill | ICT_SKILL | A categorical variable indicating level of skills of ICT staff. |
| ICT use | ICT_USE | A categorical variable measuring the proportion of staff uses ICT every day in their work. |

The level of digital transformation (DT) is a composite index constructed using three indicators: (i) the perception of the SME owners/managers and the extent to which their businesses are influenced by digital technologies, (ii) perception of opportunities generated from the use and access of digital technologies, and (iii) the perception of the extent of adoption of digital technologies. The Principal Component Analysis (PCA) is used to construct the composite index to define the level of digital transformation (DT) perceived

by owners/managers. The rationale behind constructing the index is to avoid potential multicollinearity problems. Further, the study does not use a traditional research model since the interaction effect and baseline models are used to hypothesise the determinants of DT. PCA uses the manipulation of data matrices to condense the dimensions of covariates, while at the same time maximizing the amount of variation while building a composite index is a better approach than modelling equations with separate indicators as it inherits the aggregate effect of all indicators [71]. To operationalise this alternative approach, PCA transforms the data into new variables which are not correlated.

PCA is used to construct new variables ($P_i$) from a set of variables, $X_j$ ($j$ = 1, 2, ... , $n$). These variables are referred to as Principal Components which are linear combinations of $X$'s. The following equation is used to construct the composite index of DT:

$$\text{DT} = \sum_{i=1}^{3} a_{ij} \frac{X_{ij}}{Sd(X)_i} \tag{1}$$

where DT is the composite index of the digital transformation perceived by the entrepreneurs, $Sd$ is the standard deviation, $X_{ij}$ is the $i$th variable in $j$th person, and $a_{ij}$ is the factor loading derived through PCA. In multiple linear regression models, when covariates are correlated with each other, PCA is used as one of these remedial measures.

RBV postulates that similar to the definition of resources, the capabilities of a firm refer to the individual and organizational routines and their implementation to create a competitive advantage [34]. In this connection, a firm's internal capability is defined as its capacity to integrate its capabilities and resources reflected across its use of digital technologies. In turn, this refers to a digital opportunity that a firm can witness by relating an organisation's internal capacity to its ability to adapt to changes [72]. This is the rationale behind choosing the above-mentioned three indicators to define digital transformation within the RBV framework.

### 3.2. Data Collection and Sampling Method

The data was collected from SMEs operating in the Western Downs region of Queensland, a state of Australia, with a total population of 2107 SMEs. Samples were selected through a two-stage cluster sampling approach, initially selecting four local areas at random, i.e., Chinchilla, Dalby, Murilla-Wandoan, and Tara, within the study site and then randomly selecting the designated number of SMEs (i.e., survey participants) from each location. In this study, SMEs refer to businesses that have 0 to 199 employees. The study area includes regional (town centres), rural and remote locations, which are typical of regional Australia [25]. The survey instrument was pretested with 10 participants in the study area to check the validity and appropriateness of wording, formatting, and sequencing of questions. The questions were refined based on the pilot outcomes. The final survey included 54 questions about access to, and use of, digitisation for innovation, including firm characteristics, and participants' demographics. The SMEs were contacted and asked if they preferred to participate in the survey either by telephone, mail, or online. The owner/manager of the surveyed SMEs was requested to fill out the questionnaire as they were assumed to be the key decision-makers of the respective businesses. Two hundred and eighty-one (281) firms satisfied this criterion. One hundred and fifty-nine of the total respondents were women owner-managers representing approximately 56.6 percent of the sample.

### 3.3. Estimation Strategies

#### 3.3.1. Baseline Estimation

The first research question of the study strives to explore the resource determinants of digital transformation both tangible and intangible. The selection of variables is determined by two factors. Firstly, from the review of the existing literature, it is evident that several socio-demographic factors and the adoption of digital technologies can shape the

perception of digital transformation (Table 1). Secondly, according to the RBV for sustained performance, an organisation should be equipped with different types of resources and capabilities including physical capital resources (e.g., internationalisation), technological resources (e.g., adoption of digital technology, use of digital marketing tools, research and development efforts toward innovation), human resources (e.g., organisational culture including digital attitude to work and attitudes toward women) [34,64,73,74]. In a study by Gupta et al. [75] on agile software development (ASD), where cultural awareness affects the use of social and technical agile practices, e.g., agile values, the authors found that ASD practices should reflect new cultural assumptions where firms need to identify and appropriately manage the cultural transitions involved in the adoption process of ASD practices. Culture for instance also reflects an open and positive attitude towards digital technologies that enables individuals to be better prepared when organisations consider the adoption process [76,77]. It is also evident from existing empirical studies, E-commerce driven by digital technologies opens a wider avenue for SMEs by removing the barriers arising from geographical remoteness [78]. Taking all these factors together, three baseline models have been hypothesised as follows:

$$
\begin{aligned}
DT = \alpha + \text{ECOM} + \text{CUL} + \text{INNOV} + \text{GENDER} + \text{EMP} + \text{REMOTE} + \text{INT} \\
+ \text{PROFIT} + \text{ICT\_SKILL} + \text{ICT\_USE} + u + \varepsilon
\end{aligned}
\tag{2a}
$$

$$
\begin{aligned}
DT = \alpha + \text{WSP} + \text{CUL} + \text{INNOV} + \text{GENDER} + \text{EMP} + \text{REMOTE} + \text{INT} \\
+ \text{PROFIT} + \text{ICT}_{\text{SKILL}} + \text{ICT}_{\text{USE}} + u + \varepsilon
\end{aligned}
\tag{2b}
$$

$$
\begin{aligned}
DT = \alpha + \text{SN} + \text{CUL} + \text{INNOV} + \text{GENDER} + \text{EMP} + \text{REMOTE} + \text{INT} + \text{PROFIT} \\
+ \text{ICT\_SKILL} + \text{ICT\_USE} + u + \varepsilon
\end{aligned}
\tag{2c}
$$

where $u_i$ represents the industry fixed effect, $\varepsilon_{it}$ is the error term. $\alpha$ and the vector $\Lambda$ are the parameters to be estimated. All other variables are defined as before.

To begin, the Ordinary Least Squares (OLS) is applied to all three baseline models. Breusch-Pagan test for heteroscedasticity reveals that all the models suffer from the presence of heteroskedasticity problems. Therefore, the study incorporated FGLS and GLM to account for the heteroskedasticity issue. However, the research models appeared to be free from the problem of multicollinearity. The variance inflation factor (VIF) for the explanatory variables used in OLS estimation of the three aforementioned equations is well below the threshold value of 5.

### 3.3.2. Feasible General Least Squares (FGLS) Estimation

FGLS is widely used to estimate coefficients of a regression model holding the zero-conditional mean assumption intact. FGLS estimator is a special case of GLS estimation where the errors are not known (non-IID [Independent and identically distributed]). Since IID errors are not known, the estimator is infeasible. FGLS assumes a structure that describes how the errors deviate from IID errors. Provided the assumption, IID errors can be estimated consistently. It can be argued that a robust estimator of the VCE approach can be used instead of FGLS estimators to account for non-IID disturbances. By placing more structure on the estimations, FGLS estimators yield more efficient point and consistent estimators than robust estimators of the VCE approach [79].

### 3.3.3. Generalised Linear Model (GLM) Estimation

GLM estimators are basically a generalisation of nonlinear least squares. These estimators are also appropriate for data that can potentially exhibit heteroskedasticity. Following the GLM framework advocated by Venables and Ripley [80], the GLM framework has been applied in the study using the statistical package 'Stata 14.2'. Precisely, the GLM is a development of linear models to incorporate both non-normal response distributions and transformations to linearity in a simple way. The Generalized least squares (GLS) estimates are the Maximum-Likelihood (ML) estimates for this model. When several response vari-

ables are calculated and used simultaneously in the analysis with the same explanatory variables, the combinations of a Gaussian family with 'identity' links provide a multivariate normal analysis. The Akaike's Information Criterion (AIC) was used to evaluate the best fit of the different models. The AIC values are calculated as a penalised log-likelihood with AIC = $-2 \times$ *log-likelihood* + $2(p + 1)$, where $p$ is the number of parameters in the model, and 1 is added for the predicted variance. AIC is advantageous as it explicitly penalizes any superfluous parameters in the model by including $2(p + 1)$ to the deviance. The smaller AIC indicates a better fit to the observed data in comparing two or more models.

A battery of sensitivity analysis is also conducted to cross-check the validity of the FGLS and GLS estimations of the baseline models. These robustness checks include estimation of baseline models (Equations (2a)–(2c)) using median regression. The objective of median regression is to estimate the median of the outcome variable which is conditional on the values of the explanatory variables [81]. This method is similar to OLS regression. FGLS and GLS estimations are used to answer the first research question.

### 3.3.4. Estimation of the Interaction Effect of Gender

To answer the second research question, it has been explored whether the gender of an SME's owner/manager has an impact on the drivers of digital transformation. In the baseline models, the use of E-commerce, websites, and social network platforms is assumed as major drivers of digital transformation perceived by the business owners/managers. The possible course of interaction of gender with respect to digital marketing has been outlined in Equations (3a),(3b). However, in order to arrive at a meaningful result from the interaction effects, the independent variables (ECOM, WSP, and SN), as outlined below, need to be centralised first. The rationale is that once an interaction effect is added to the model, the main effects may or may not be interesting. Therefore, the results may not be meaningful in that case [82]. To overcome this limitation, the independent variables are centred first by subtracting the mean from each case, and then computing the interaction term and thereafter estimating the model [83]. GLM estimators are used to estimate the coefficients of the explanatory variables of the following equations. Here, the decision rule is that if the standardised regression coefficient of the interaction term is statistically significant, and at the same time, the standardised regression coefficient of the moderating variable is not significant, then that particular moderating variable is said to have a significant impact on the dependent variable [82]. Therefore, the three interaction effect models are as follows:

$$
\begin{aligned}
DT = \alpha\ &+\ \text{ECOM} + \text{CUL} + \text{INNOV} + \text{GENDER} + \text{GENDER}_{\text{ECOM}} + \text{EMP} + \text{REMOTE} \\
&+\ \text{INT} + \text{PROFIT} + \text{ICT}_{\text{SKILL}} + \text{ICT}_{\text{USE}} + u + \varepsilon
\end{aligned}
\tag{3a}
$$

$$
\begin{aligned}
DT = \alpha\ &+\ \text{WSP} + \text{CUL} + \text{INNOV} + \text{GENDER} + \text{GENDER\_WSP} + \text{EMP} + \text{REMOTE} \\
&+\ \text{INT} + \text{PROFIT} + \text{ICT\_SKILL} + \text{ICT\_USE} + u + \varepsilon
\end{aligned}
\tag{3b}
$$

$$
\begin{aligned}
DT = \alpha\ &+\ \text{SN} + \text{CUL} + \text{INNOV} + \text{GENDER} + \text{GENDER}_{\text{SN}} + \text{EMP} + \text{REMOTE} + \text{INT} \\
&+\ \text{PROFIT} + \text{ICT}_{\text{SKILL}} + \text{ICT}_{\text{USE}} + u + \varepsilon
\end{aligned}
\tag{3c}
$$

where, GENDER_ECOM, GENDER_WSP, and GENDER_SN denote the interaction effect of gender on the use of E-commerce applications, websites, and social network platforms, respectively. Likewise, the baseline equations, ECOM, CUL, INNOV, EMP, REMOTE, INT, PROFIT, ICT_SKILL, and ICT_USE are the control variables.

Likewise, the baseline estimations, a battery of robustness checks are also conducted to cross-validate the GLS estimations of the interaction effects models. These robustness checks are conducted using median regression for the interaction effect models (Equations (3a)–(3c)). In addition, to explore the potential differences in entrepreneurs' perception of the digital transformation and the use of the social network for business purposes between males and females, the Blinder–Oaxaca decomposition method is applied using the Stata 14.2 Oaxaca routine. This method was proposed by Blinder [84] and Oaxaca [85] for estimating gender-based wage discrimination.

## 4. Empirical Results

### 4.1. Descriptive Statistics

Descriptive statistics of the variables of the entire sample as well as for the two groups of samples, namely female and male-led enterprises, are presented in Table 2. It also lists the results of the independent sample *t*-test. The mean perceived scores of the composite index of DT for female- and male-led SMEs are 3.75 and 3.74, respectively. The high perceived score of digital transformation indicates that SMEs operating in regional areas perceive the use of digital transformation as a means of using strategic resources and capabilities that create digital opportunities for businesses.

**Table 2.** Descriptive statistics and independent sample *t*-test.

| Variable | All | | Male | | Female | | *t*-Statistic |
|---|---|---|---|---|---|---|---|
| | Mean | Std. | Mean | Std. | Mean | Std. | |
| DT | 3.7450 | 0.7352 | 3.7354 | 0.8090 | 3.7524 | 0.6758 | −0.1923 |
| ECOM | 0.3950 | 0.7897 | 0.4098 | 0.4938 | 0.3836 | 0.4878 | 0.4437 |
| WSP | 0.0530 | 0.5008 | 0.5327 | 0.5009 | 0.4842 | 0.5013 | 0.8042 |
| SN | 0.9039 | 0.2952 | 0.8606 | 0.3477 | 0.9371 | 0.2435 | −2.1656 ** |
| CUL | 3.8398 | 0.8573 | 3.8606 | 0.8461 | 3.8238 | 0.8681 | 0.3557 |
| INNOV | 2.9624 | 0.5078 | 2.9555 | 0.5508 | 2.9677 | 0.4739 | −0.1997 |
| EMP | 9.8967 | 21.1111 | 10.6885 | 22.1606 | 9.2893 | 20.3190 | 0.5500 |
| REMOTE | 0.2241 | 0.7671 | 0.1721 | 0.3790 | 0.2641 | 0.4422 | −1.8377 *** |
| INT | 1.2811 | 0.5655 | 1.2459 | 0.7639 | 1.3081 | 0.7709 | −0.6738 |
| PROFIT | 2.2419 | 0.8246 | 2.2622 | 0.5574 | 2.2264 | 0.5728 | 0.5265 |
| ICT_SKILL | 3.0960 | 0.8246 | 2.9426 | 0.9118 | 3.2138 | 0.7322 | −2.7648 * |
| ICT_USE | 2.8861 | 1.4222 | 3.0245 | 1.1469 | 2.7798 | 1.4215 | 1.4323 |
| Observations | 281 | | 122 | | 159 | | |

Note: Figures in the parentheses represent standard error. *, ** and *** denote statistically significant at 1 percent, 5 percent and 10 percent, respectively.

The independent sample *t*-test reveals that a significant gender difference persists in the use of social networks for businesses. The use of social media networks is significantly higher among female-led enterprises compared to their male-led counterparts which confirms previous research noted earlier [70]. Similarly, the presence of specialised or ICT-skilled staff is significantly higher among female-led enterprises than male-led counterparts. The results of the nonparametric test, namely the Mann-Whitney U test corroborate the results obtained through the independent sample *t*-test (for brevity, these statistics are not reported here). Overall, these preliminary results indicate that there is a significant difference between female-led and male-led SMEs with regard to the digital technology-related resources and capabilities of sample firms, viz., use of social network platforms, and ICT-skilled staff.

### 4.2. Results of Baseline Estimation

Table 3 presents the results of the baseline estimations using FGLS based on Equations (3a)–(3c). For each equation, two models are estimated, the first one is without industry fixed effects and the second one is with industry fixed effects. The industry fixed effect model controls for potential industry effects. In Models 1 and 2, workplace culture, innovation, and remoteness are found to be significant predictors of digital transformation as perceived by the business owners/managers (columns 1 and 2). As per these results, the use of E-commerce applications has no significant influence on the perception of digital transformation. Looking at the results of Models 3 and 4, it is apparent that web pres-

ence along with workplace culture, innovation, remoteness, and ICT use are significant predictors of the digital transformation of SMEs (columns 3 and 4). By the same token, considering Models 5 and 6, the use of social network platforms, workplace culture, innovation, remoteness, and ICT use significantly explain digital transformation (columns 5 and 6).

**Table 3.** Estimation results of the baseline models using FGLS.

| Variable | Equation (2a) | | Equation (2b) | | Equation (2c) | |
|---|---|---|---|---|---|---|
| | **(1)** | **(2)** | **(3)** | **(4)** | **(5)** | **(6)** |
| **ECOM** | 0.0001 | 0.0123 | | | | |
| | (0.0436) | (0.0404) | | | | |
| **WSP** | | | 0.0646 *** | 0.0817 ** | | |
| | | | (0.0400) | (0.0403) | | |
| **SN** | | | | | 0.4202 * | 0.4222 * |
| | | | | | (0.1217) | (0.1251) |
| **CUL** | 0.2237 * | 0.2356 * | 0.2251 * | 0.2292 * | 0.2318 * | 0.2368 * |
| | (0.0224) | (0.0200) | (0.0217) | (0.0207) | (0.0206) | (0.0211) |
| **INNOV** | 0.9997 * | 1.0270 * | 0.9915 * | 0.9980 * | 1.023 * | 1.2014 * |
| | (0.0483) | (0.0450) | (0.0449) | (0.0435) | (0.0405) | (0.0406) |
| **GENDER** | −0.5665 | −0.0394 | −0.0517 | −0.0404 | −0.0464 | −0.0446 |
| | (0.0398) | (0.0358) | (0.0377) | (0.0346) | (0.0349) | (0.0348) |
| **EMP** | −0.0003 | 0.0001 | <−0.0001 | −0.0006 | 0.0014 | 0.0015 |
| | (0.0015) | (0.0013) | (0.0012) | (0.0015) | (0.0364) | (0.0013) |
| **REMOTE** | 0.1133 * | 0.1164 * | 0.1281 * | 0.1157 | 0.0956 * | 0.0996 * |
| | (0.0408) | (0.0375) | (0.0387) | (0.0383) | (0.0364) | (0.0371) |
| **INT** | 0.0184 | 0.0117 | 0.0143 | −0.0071 | 0.0035 | 0.0074 |
| | (0.0323) | (0.0312) | (0.0344) | (0.0333) | (0.0304) | (0.0337) |
| **PROFIT** | 0.0024 | 0.0024 | −0.0023 | 0.0084 | −0.0225 | −0.0241 |
| | (0.0359) | (0.0358) | (0.0340) | (0.0347) | (0.0310) | (0.0330) |
| **ICT_SKILL** | 0.0460 | 0.0326 | 0.0480 | 0.0434 | 0.0402 | 0.0431 |
| | (0.0352) | (0.0331) | (0.0336) | (0.0329) | (0.0312) | (0.0330) |
| **ICT_USE** | 0.0259 | 0.0171 | 0.0254 | 0.0264 *** | 0.0240 | 0.0215 *** |
| | (0.0161) | (0.0157) | (0.0159) | (0.0159) | (0.0151) | (0.0151) |
| **Constant** | −0.2972 *** | −0.3627 ** | −0.3062 ** | −0.3445 ** | −0.6993 * | −0.7611 |
| | (0.1579) | (0.1532) | (0.1530) | (0.1487) | (0.1833) | (0.1922) |
| **Industry FE** | No | Yes | No | Yes | No | Yes |
| **Observations** | 281 | 281 | 281 | 281 | 281 | 281 |
| **R−squared** | 0.8136 | 0.8414 | 0.8329 | 0.8445 | 0.8530 | 0.8556 |

Note: Figures in the parentheses represent standard error. *, ** and *** denote statistically significant at 1 percent, 5 percent and 10 percent, respectively.

Similar results have been found when the above six models are re-estimated using GLM (Table 4). Overall, the use of social network platforms, workplace culture, innovation, remoteness, and ICT use, significantly explain business owner/managers' perceptions of digital transformation.

**Table 4.** Estimation results of the baseline models using GLM.

| Variable | Equation (3a) | | Equation (3b) | | Equation (3c) | |
|---|---|---|---|---|---|---|
| | **(1)** | **(2)** | **(3)** | **(4)** | **(5)** | **(6)** |
| **ECOM** | 0.0440 | 0.0266 | | | | |
| | (0.0619) | (0.0633) | | | | |
| **WSP** | | | 0.0850 | 0.1101 *** | | |
| | | | (0.0601) | (0.0626) | | |
| **SN** | | | | | 0.4934 * | 0.4857 * |
| | | | | | (0.0924) | (0.0924) |
| **CUL** | 0.0210 * | 0.2059 * | 0.2029 * | 0.2052 * | 0.2134 * | 0.2149 * |
| | (0.0358) | (0.0360) * | (0.0351) | (0.0350) | (0.0335) | (0.0335) |
| **INNOV** | 0.8367 * | 0.8461 | 0.8365 * | 0.8381 * | 0.8611 * | 0.8644 * |
| | (0.0672) | (0.0675) | (0.0650) | (0.0648) | (0.0614) | (0.0614) |
| **GENDER** | −0.0045 | −0.0051 | −0.0054 | −0.0050 | −0.4060 | −0.0452 |
| | (0.0567) | (0.0567) | (0.0565) | (0.0563) | (0.0544) | (0.0544) |
| **EMP** | 0.0011 | 0.0011 | 0.0009 | 0.0008 | 0.0006 | 0.0006 |
| | (0.0014) | (0.0014) | (0.0014) | (0.0014) | (0.0014) | (0.0014) |
| **REMOTE** | 0.1283 *** | 0.1111 | 0.01386 ** | 0.1118 | 0.0926 | 0.0738 |
| | (0.0690) | (0.0725) | (0.0691) | (0.0721) | (0.0694) | (0.0694) |
| **INT** | 0.0662 ** | 0.0405 | 0.0709 *** | 0.0498 | 0.1077 ** | 0.0939 ** |
| | (0.0386) | (0.0405) | (0.0386) | (0.0402) | (0.0394) | (0.0394) |
| **PROFIT** | 0.0016 | 0.0094 | −0.0006 | 0.0064 | −0.0273 | −0.0183 |
| | (0.0523) | (0.0529) | (0.0519) | (0.0523) | (0.0503) | (0.0503) |
| **ICT_SKILL** | 0.0824 ** | 0.0812 ** | 0.0805 ** | 0.0357 ** | 0.0860 ** | 0.0842 |
| | (0.0358) | (0.0358) | (0.0357) | (0.0357) | (0.0341) | (0.0341) |
| **ICT_USE** | 0.0681 * | 0.0688 * | 0.0628 * | 0.0618 * | 0.0651 * | 0.0654 * |
| | (0.0226) | (0.0226) | (0.0228) | (0.0228) | (0.0215) | (0.0215) |
| **Constant** | 0.2340 | −0.1682 | −0.1135 | −0.1722 | −0.6323 | −0.6323 * |
| | (0.2340) | (0.2384) | (0.0228) | (0.2302) | (0.2355) | (0.2355) |
| **Industry FE** | No | Yes | No | Yes | No | Yes |
| **Observations** | 281 | 281 | 281 | 281 | 281 | 281 |
| **Log likelihood** | −137.9901 | −172.8495 | −173.21 | −173.33 | −159.8577 | −159.1563 |
| **AIC** | 1.3166 | 1.3227 | 1.3111 | 1.3119 | 1.2160 | 1.2253 |

Note: Figures in the parentheses represent standard error. *, ** and *** denote statistically significant at 1 percent, 5 percent and 10 percent, respectively.

### 4.3. Interaction Effect of Gender

Table 5 reports the moderation effect of gender on the impact of the use of digital marketing tools for business. GLS estimators are used to examine the interaction effect of gender. It is evident from the results that gender has significantly moderated the effect of the use of social networking on the perception of the digital transformation of SMEs.

**Table 5.** Estimation results of the interaction effects of gender using GLM.

| Variable | Equation (2a) | Equation (2b) | Equation (2c) |
|---|---|---|---|
| | (1) | (2) | (3) |
| ECOM | 0.0667 | | |
| | (0.0929) | | |
| WSP | | 0.1040 | |
| | | (0.0896) | |
| SN | | | 0.2910 ** |
| | | | (0.1187) |
| CUL | 0.2079 * | 0.2051 * | 0.2110 * |
| | (0.0362) | (0.0351) | (0.0332) |
| INNOV | 0.8398 * | 0.8387 * | 0.8646 * |
| | (0.0684) | (0.0652) | (0.0608) |
| GENDER | −0.0048 | −0.0049 | −0.4441 |
| | (0.0567) | (.0564) | (0.0539) |
| GENDER_ECOM | −0.0685 | | |
| | (0.1160) | 0.0108 | |
| GENDER_WSP | | (0.1150) | |
| GENDER_SN | | | 0.4792 ** |
| | | | (0.1865) |
| EMP | 0.0011 | 0.0007 | 0.0008 |
| | (0.0014) | (0.0014) | (0.0014) |
| REMOTE | 0.1065 | 0.1116 | 0.0684 |
| | (0.0730) | (0.0723) | (0.0687) |
| INT | 0.0509 | 0.0496 | 0.0745 *** |
| | (0.0405) | (0.0403) | (0.0397) |
| PROFIT | 0.0085 | 0.0058 | −0.0300 |
| | (0.0530) | (0.0527) | (0.0500) |
| ICT_SKILL | 0.0826 ** | 0.0823 | 0.0849 ** |
| | (0.0360) | (0.0367) ** | (0.0338) |
| ICT_USE | 0.0683 * | 0.0618* | 0.0687 * |
| | (0.0226) | (0.0229) | (0.0743) |
| Constant | −0.1746 | −0.1666 | 0.0399 *** |
| | (0.2389) | (0.2380) | (0.2483) |
| Industry FE | Yes | Yes | Yes |
| Observations | 281 | 281 | 281 |
| Log likelihood | −172.6660 | −171.3256 | −155.7266 |
| AIC | 1.3285 | 1.31 | 1.2080 |

Note: Figures in the parentheses represent standard error. *, ** and *** denote statistically significant at 1 percent, 5 percent and 10 percent, respectively.

The coefficient (0.4792) of the interaction variable (SN multiplied by GENDER) is statistically significant at a 5 percent level while the coefficient for the moderating variable (GENDER) is not significant (column 3). In addition, the use of social network platforms,

workplace culture, innovation, internationalisation, ICT skills, and ICT use have a significant impact on the perception of digital transformation.

*4.4. Robustness Checks*

Table S1 in the supplementary materials provides a series of sensitivity analyses for baseline models using median regression for three Equations 2a to 2c. It reveals that the use of digital platforms significantly explained the perceived digital transformation. These results are consistent with the baseline estimation using FGLS and GLM reported earlier (Tables 3 and 4). Table S2 summarises the results from a series of sensitivity analyses for the interaction effect models using median regression. It reconfirms the findings of basic interaction models. Likewise, in the main interaction models, it is evident from the results that gender has significantly moderated the effect of the use of social networking on the perceived score of digital transformation in the median regression.

The results obtained from the Blinder–Oaxaca decomposition are summarised in Table S3 in the supplementary materials. These results indicate that the perceived digital opportunity of a female-led SME is higher than that of a male-led SME; however, this difference is not statistically significant. The use of social networks for female-led businesses is significantly higher than that of male counterparts.

**5. Discussion**

Prior to the current study, little was known about the perceived role of digital transformation in rural businesses as a viable resource. The authors noted the potential importance of these differences given prior knowledge about digital inequality and unequal access to digital technologies particularly in regional communities [27], where women owners and managers were expected to fare worse than their male counterparts. Scholars were thus no wiser about the extent to which digital transformation was occurring among SMEs in rural communities.

Several interesting findings have emerged from the results of this study. The most obvious finding is that the adoption of digital marketing tools has a positive impact on the perceived digital transformation of SMEs. To be specific, the use of social networking platforms and web presence significantly influence the perception of digital transformation of SMEs. These findings are consistent with earlier research which showed that the use of social networks and web presence for business purposes has a positive effect on the digital transformation of SMEs. Digital transformation is achieved through online marketing and easing the communication process between buyers and sellers [17,32,48,53]. Surprisingly, the effect of E-commerce adoption on perceived digital transformation was found to be statistically insignificant. One plausible explanation behind this unexpected finding is the comparatively lower rate of adoption of E-commerce applications by SMEs in regional areas. The results indicate that more than 60 percent of the SMEs in the study sample do not use E-commerce platforms to sell their goods and services.

Another finding of the study reveals that the perceived benefits of digital transformation are positively associated with ICT use. This finding is corroborated by the existing literature [45,86]. Among other enterprise-specific characteristics, workplace culture, innovation activities, and internationalisation have demonstrated a significant positive association with the level of digital transformation. These findings broadly support previous empirical studies [13,41]. Moreover, the disruptive nature of COVID-19 has created a spectrum of opportunities for innovation and transformation in business enterprises. It is evident that during the pandemic, business owners adapt to the emergency conditions and pivot their business models to seize opportunities offering new products/services and marketing differently. Such observations are in line with the findings of Baig et al. [5] and Kamal [87]. As digital transformations are heterogeneous and are always evolving particularly during a pandemic period, SMEs need to acquire a high level of leadership capabilities, upskilled digital skills, and sufficient business domain knowledge that can help them cope with the associated challenges [17].

Interestingly, a few other enterprise-specific features, for example, size, profitability, and digital skills, are reported to have no significant impact on the perception of digital transformation. These findings are not consistent with the evidence from the existing literature [88–91]. These discrepancies may be due to the adoption of advanced methodological approaches in this study to tackle the problem of multicollinearity, and the heteroskedasticity problem, which was overlooked in existing empirical studies.

Consistent with the literature, this study finds that there is a significant difference between female- and male-led businesses regarding their perception of digital transformation. Specifically, gender has significantly moderated the effect of the use of social networking on the perception of digital transformation. However, the findings cannot account for the social, cultural, and economic contextual effects on women-led business in regional Australia. This result postulates that even though women-owned/managed SMEs use social media to a greater extent than their male counterparts, the skill levels of ICT staff in women-owned/managed SMEs are greater than in male-owned/managed SMEs. The latter may be an indication that women need more assistance with digital strategies than their male counterparts. Research has also shown that males are more likely to seek 'technical' roles whereas women are less likely to engage in IT skill building which is evident from their tendency to pursue 'softer' technology roles [33,92]. In the context of this study, clearly women recognise the importance of and engaging in social media to build their businesses. One could argue that they do so because they value relationship management to build their businesses, however, they employ ICT staff to a greater extent than male-owned businesses because they may not have ICT skills themselves to drive digital transformation in their businesses. More research is therefore required about gender inequality including the social, cultural, and economic reality of female-led business in regional Australia. It is particularly important in times of a national and worldwide crisis, to assess whether any gender digital inequality exist between male and female-led SMEs during a pandemic shock recovery.

In addition, this study makes a major contribution by extending existing literature related to digital technology use and application of SME owners/managers located within regional locations. Prior to the current study, little was known about the perceived role of digital transformation in rural communities and how perceptions of digital transformation vary between female-led and male-led businesses. Given the strong moderating effect of gender on these perceptions, this is a significant finding. This finding also indicates that within social, cultural, political, and economic domains, new digital technology strategies need to be considered within rural enterprises and local government jurisdictions. Furthermore, new strategies related to digital gender inequality within regional areas need to be formulated to help close the gaps regarding digital transformation.

## 6. Conclusions and Policy Implications

This study examined the factors determining SME owner/managers' perception of digital transformation in their business within a regional context. This relationship was further analysed from a gender inequality perspective. It was found that digital technology resources and capabilities such as the use of social networks, workplace culture, innovation, internationalisation, and ICT use significantly and positively contribute to sustainable digital transformation. In addition, the gender of business owners/managers is reported to significantly moderate the impact of the use of social networking on digital transformation.

The present study makes some important contributions to policy and the existing literature for Australian regional SMEs and may also have implications for policy in other regional areas around the world. First, the findings suggest the need for policymakers to pay attention to the gender composition and levels of digital literacy of regional SME owners/managers, including that of their staff, to ease the digital disruption in regional areas. Second, despite the possible opportunities that SMEs can explore resulting from digital transformation, the rapid expansion of digital technologies among regional SMEs disrupts the existing way of doing business which tend to pose risks of higher gender

inequality. Third, the Australian government should carefully consider both the opportunities and threats that characterise digital disruption, which mostly occur due to digital transformation interventions to boost regional businesses. Fourth, cyber security and privacy are very important factors worth considering when designing programs and policies aimed at optimising and maximising digital transformation for businesses. Fifth, at the National, State, and Regional Council levels, Governments should plan to oversee and advise businesses of any potential privacy threats that may emerge from the adoption of conventional digital technologies.

Finally, it is evident from recent research that the educational attainment of female entrepreneurs strengthened the positive impact of digital transformation on the financial performance of SMEs during the COVID-19 pandemic [32]. Therefore, targeted training for women business owners/managers could enhance their critical thinking and communication skills further, which are indispensable for decision-making during periods of crisis such as COVID-19.

## 7. Limitation and Directions for Future Research

This study has several limitations that could represent avenues for future research. Further research could be conducted on a larger sample of SMEs across Australia to examine the factors associated with digital transformation. The results of the study were influenced by the choice of methodology, including the survey and econometric models used. To benefit from the triangulation of the data in future research, future studies might also include focus groups of female and male SME managers, providing greater insights as to why the determinants identified in the present study are significant within the context of digital transformation in regional communities. Also, it may be useful in future research to apply a social constructionist perspective and critical realist ontology, enabling rich data to be collected in one-on-one interviews. Accordingly, a mixed-methods approach may be considered within the context of exploring the real reasons for the digital divide within regional communities. Moreover, a different mix of methodologies may help to extend the current results and enable a better understanding of pandemic-related factors that influence specific demographic groups. The current study placed emphasis only on the opportunities driven by digital technologies in SMEs. Future research could be conducted on the threats posed by the digital transformation of business enterprises.

**Supplementary Materials:** The following are available online at https://www.mdpi.com/article/10.3390/su14010535/s1, Table S1: Robustness checks of baseline models, Table S2: Robustness checks of the interaction effects of gender, Table S3: Decomposition of the differences in DDO and use of the social network for business purposes by gender.

**Author Contributions:** K.A. designed the study. M.A.A. and K.A. conducted data cleaning and analysis and wrote the first draft of the paper. All authors provided substantial contributions to subsequent drafts and approved the final version of the manuscript for publication. All authors have read and agreed to the published version of the manuscript.

**Funding:** This research was funded by the Western Downs Regional Council and the Australian Government's Collaborative Research Networks program.

**Institutional Review Board Statement:** The study was conducted in accordance with the Declaration of Helsinki, and the ethical approval for this study was obtained from the University of Southern Queensland Human Research Ethics Committee (Approval Number: H13REA150, and date of approval: 31 July 2013).

**Informed Consent Statement:** Informed consent was obtained from all subjects involved in the study.

**Data Availability Statement:** The data presented in this study are available on request from the corresponding author.

**Acknowledgments:** This project was supported by the Western Downs Regional Council and the Australian Government's Collaborative Research Networks program awarded to the first author at the University of Southern Queensland. The first author thanks those who participated by completing the survey questionnaires; special thanks to Mohammad Shahiduzzaman for assisting in data collection and Market Facts Pty Ltd. in Queensland for conducting the survey. All authors gratefully acknowledge the comments of reviewers and editors of this manuscript.

**Conflicts of Interest:** The authors indicate no conflict of interest.

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
