# Peer review of "Digital Transformation among SMEs: Does Gender Matter?"

_sustainability, doi:10.3390/su14010535_

Round 1

Reviewer 1 Report

The article presents in an interesting perspective the issues related to

 Digital transformation among small and medium enterprises in 1 regional Australia.

Strengths

The article presents the empirical data and detailed analyzes.

Weaknesses

The timeliness and significance of the problem should be broadened.

The methodology of empirical research should be described in more detail.

The technique of selecting the research sample should be clarified.

Conclusions should be expanded, pointing to the limitations of the analyzed problem and defining the directions of further research.

Authors should:

- define the essence and importance of the described problem;

-detail the methodology of the conducted empirical research;

-justify the scope of the analyzed research sample;

-pointing to the limitations of the analyzed problem and

defining the directions of further research.

Author Response

The authors would like to thank reviewer 1 for their insightful comments in relation to the manuscript. Please see attached responses.

Reviewer 2 Report

This is a timely and interesting topic.

Although the topic captures the reader's attention, the article seems incomplete and has some shortcomings.

One of the most relevant limitations is the authors' option to present a generic literature review, without presenting a strong conceptual model.

In relation to the literature review, the authors benefit from the fact that the digital transformation is a recent topic, so the bibliographical references are very current and from quality journals. However, most bibliographical references seem to be incomplete, for example, see the lines:

  1. Chege, Wang, , (2020)
  2. Rotar et al (2019),
  3. Ramayah et al. (2016).

I suggest that you complete all references and do a second review before resubmitting a new version of the article.

Within the scope of the literature review, I believe that it may be relevant to better justify the option of digital transformation, resorting to its distinction from similar terms that are sometimes still used interchangeably, such as: digitalization, digitation or digital disruption. I would recommend the Verhoef et al. (2021) article.

Verhoef, P. C., Broekhuizen, T., Bart, Y., Bhattacharya, A., Dong, J. Q., Fabian, N., & Haenlein, M. (2021). Digital transformation: A multidisciplinary reflection and research agenda. Journal of Business Research122, 889-901.

In terms of generalization, I would like to know how the results could be generalized to other parts of the globe (e.g., Asia). Or at least that the article presented some contributions in that regard, not being so focused on a particular region of the globe.

These are some issues that I would like to see better defined and that I think would improve the article.

Author Response

The authors would like to thank reviewer 2 for their insightful comments of the manuscript. Please see attached response.

Reviewer 3 Report

Dear authors,

This manuscript was very interesting. I just recommend adding some information in part 3.2. When did the research? How was collected data?

sincerely

Author Response

The authors would like to thank the reviewer for their insightful comment of the manuscript. Please see attached responses.

Reviewer 4 Report

Comments to the Author
The paper investigated the digital transformation among small and medium enterprises. The topic is exciting and worthy of investigation. However, I think the following comments will make it better and more robust.
* The title can be made short by using abbreviation e.g., "Digital transformation among SMEs: Does gender matter?".
*The researcher(s) should pay attention to the research gap that is still not sufficient. Therefore, please add more arguments related to the research gap in the introduction.
*Please add one or two paragraphs on the impact of the COVID-19 pandemic on small and medium enterprises. Therefore, I think more elaboration is needed to explain the COVID-19 pandemic impact in the introduction part.
*Also, the Theoretical construct and development of hypotheses part were prepared well. Yet, still very long and needs to be minimized.
*Please use the abbreviation once being introduced. For instance, SMEs for (small and medium enterprises).
*Some citations of the reference lists are missing. Similar issues need to be fixed in reference lists. Consequently, please pay attention to the references to check them appropriately.
*Please try to restructure the research conclusion and policy implications in a separate section on research implications. The current writing is not well structured.
*Along the same lines, it is necessary to mention the research limitations and recommendations in a separate section. 
*Finally, please improve the language of the research paper.

I hope that my comments can help you to improve your manuscript.

Author Response

The authors would like to thank the reviewer for their insightful comments of the manuscript. Please see attached responses.

Round 2

Reviewer 1 Report

The new version of the manuscript contains mostly additions and corrections.

Reviewer 2 Report

The authors adequately answered to my questions.